# Semi-Supervised Learning with Declaratively Specified Entropy Constraints

**Haitian Sun**
Machine Learning Department
Carnegie Mellon University
Pittsburgh, PA 15213
haitians@cs.cmu.edu

**Lidong Bing**[*]
R&D Center Singapore
Machine Intelligence Technology
Alibaba DAMO Academy
l.bing@alibaba-inc.com

**William W. Cohen**
Machine Learning Department
Carnegie Mellon University
Pittsburgh, PA 15213
wcohen@cs.cmu.edu

## Abstract

We propose a technique for declaratively specifying strategies for semi-supervised learning (SSL). SSL methods based on different assumptions perform differently on different tasks, which leads to difficulties applying them in practice. In this paper, we propose to use entropy to unify many types of constraints. Our method can be used to easily specify ensembles of semi-supervised learners, as well as agreement constraints and entropic regularization constraints between these learners, and can be used to model both well-known heuristics such as co-training, and novel domain-specific heuristics. Besides, our model is flexible as to the underlying learning mechanism. Compared to prior frameworks for specifying SSL techniques, our technique achieves consistent improvements on a suite of well-studied SSL benchmarks, and obtains a new state-of-the-art result on a difficult relation extraction task.

## 1 Introduction

Many semi-supervised learning (SSL) methods are based on regularizers which impose "soft constraints" on how the learned classifier will behave on unlabeled data. For example, logistic regression with entropy regularization [11] and transductive SVMs [13] constrain the classifier to make confident predictions at unlabeled points; the NELL system [7] imposes consistency constraints based on an ontology of types and relations; and graph-based SSL approaches require that the instances associated with the endpoints of an edge have similar labels [31, 1, 25] or embedded representations [28, 30, 14]. Certain other weakly-supervised methods also can be viewed as constraining predictions made by a classifier: for instance, in distantly-supervised information extraction, a useful constraint requires that the classifier, when applied to the set $S$ of mentions of an entity pair that is a member of relation $r$, classifies at least one mention in $S$ as a positive instance of $r$ [12].

Unfortunately, although many specific SSL constraints have been proposed, there is little consensus as to which constraint should be used in which setting, so determining which SSL approach to use on a specific task remains an experimental question—and one which requires substantial effort to answer, since different SSL strategies are often embedded in different learning systems. To address

---

[*] This work was mainly done when Lidong Bing was working at Tencent AI Lab.

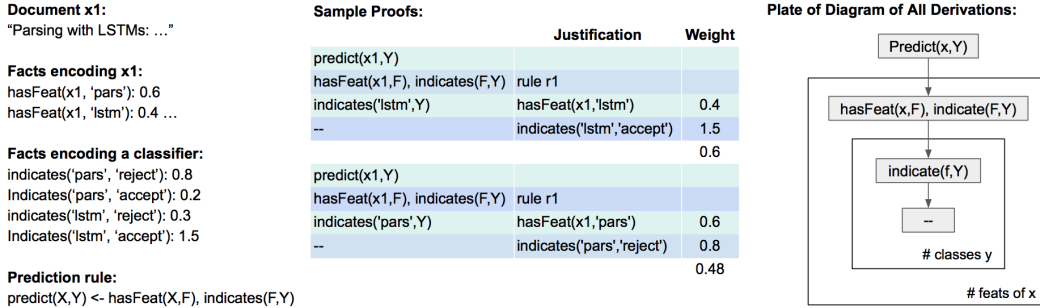

Figure 1: Specifying supervised text classification declaratively with TensorLog.

this problem, Bing et al. [3] proposed a succinct declarative language for specifying semi-supervised learners, the *D-Learner*. The D-Learner allowed many constraints to be specified easily, and allows all constraints to be easily combined and compared. The relative weights of different SSL heuristics can also be tuned collectively un the D-earner using Bayesian optimization. The D-learner was demonstrated by encoding a number of intuitive problem-specific SSL constraints, and was able to achieve significant improvements over the state-of-the-art weakly supervised learners on two real-world relation extraction tasks.

A disadvantage of the D-Learner is that it is limited to specifying constraints on, and ensembles of, a single learning system—a particular variety of supervised learner based on the ProPPR probabilistic logic [27]. In this paper we introduce a variant of the D-Learner called the DCE-Learner (for Declaratively Constrained Entropy) that uses a more constrained specification language, but is paired with a more effective and more flexible underlying learning system. This leads to consistent improvements over the original D-Learner on five benchmark problems.

In the next section, we first introduce several SSL constraints in a uniform notation. Then, in Section 3, we experiment with some benchmark text categorization tasks, to illustrate the effectiveness of the constraints. Finally, in Section 4, we generalize our model to a difficult relation extraction task in drug and disease domains, where we obtain a state-of-the-art results using this framework.

## 2 Declaratively Specifying SSL Algorithms

### 2.1 An Example: Supervised Classification

A method for declaratively specifying constrained learners with the probabilistic logic ProPPR has been previously described in [27]. Here we present a modification of this approach for probabilistic logic TensorLog [8]. TensorLog is a somewhat more restricted logic, but has the advantage that queries in TensorLog can be compiled to computation graphs in deep-learning platforms like Tensorflow. This leads to a number of advantages, in particular the ability to exploit GPU processors, well-tuned gradient-descent optimizers, and extenive libraries of learners.

We begin with the example of supervised learning for document classifiers shown in Figure 1. A TensorLog program consists of (1) a set of *rules* (function-free Horn clauses) of the form "$A \leftarrow B_1, \ldots, B_k$", where $A$ and the $B_i$'s are of the form $r(X, Y)$ and $X, Y$ are variables; and (2) a set of weighted facts, where "facts" are simply triples $r(a, b)$ where $r$ is a relation, and $a, b$ are constants. We say that constant $a$ is the "head" entity for $r$, and $b$ is the "tail" entity.

To encode a text classification task, a document $x_i$, which is normally stored as a set of weighted terms, will be encoded as weighted facts for the `hasFeature` relation. For instance, the document $x_1$ containing the text "Parsing with LSTMs" might be stoplisted, stemmed, and then encoded with the facts `hasFeature(x1,pars)` and `hasfeature(x1,lstm)`, which have weights 0.6 and 0.4 respectively (representing importance in the original document). A classifier will be encoded by a set of triples which associate features and classes: for instance, `indicates(pars,accept)` with weight 0.2 might indicate a weak association between the term `pars` and the class `accept`. Finally the classifier is the single rule

$$\text{predict(X,Y)} \leftarrow \text{hasFeature(X,F), indicates(F,Y)}$$

In TensorLog, capital letters are universally quantified variables, and the comma between the two predicates on the right-hand side of the rule stands for conjunction, so this can be read as asserting "for all $X, F, Y$, if $X$ contains a feature $F$ which indicates a class $Y$ then we predict that $X$ has the class $Y$."

Given rules and facts, a traditional non-probabilistic logic will answer a query such as `predict(x1,Y)` by finding all $y$ such that `predict(x1,Y)` can be proved, where a proof is similar to a context-free grammar derivation: at each stage one can apply a rule—e.g., to replace the query `predict(x1,Y)` with the conjunction `hasFeature(x1,f1),indicates(f1,Y)`—or one can match against a fact. The middle part of Figure 1 shows two sample proofs, which will support predicting both the class "accept" and the class "reject".

In a TensorLog, however, each proof is associated with a weight, which is simply the product of the weight of each fact used to support the proof: i.e., the weight for a proof is $\prod_{i=1}^{n} w_i$ where $w_i$ is the weight of fact used at step $i$ in the proof (or else $w_i = 1$, if a rule was used at step $i$.) TensorLog will compute *all* proofs that support *any* answer to a query, and use the sum of those weights as a confidence in an answer. Figure 1 also shows a plate-like diagram indicating the set of proofs that would be explored for this task (with the box labeled with "−" denoting a completed proof.) TensorLog aggregates the weights using dynamic programming, specifically by unrolling the message-passing steps of belief propagation on a certain factor graph into a computational graph, which can then be compiled into TensorFlow. For more details on TensorLog's semantics, see [8].

TensorLog learns to answer queries of the form $q_i(x_i, Y)$: that is, given such a query it learns to find all the constants $y_{ij}$'s such that $q_i(x_i, y_{ij})$ is provable, and associates an appropriate confidence score with each such prediction. By default TensorLog assumes there is one correct label $y_i$ for each query entity $x_i$, and the vector of weighted proof counts for the provable $y_{ij}$'s is passed through a softmax to obtain a distribution. Learning in TensorLog uses a set of training examples $\{(q_i, x_i, y_i)\}_i$ and gradient descent to optimize cross-entropy loss on the data, allowing the weights of some user-specified set of facts to vary in the optimization. In this example, if only the weights of the `indicates` facts vary, the learning algorithm shown in the figure is closely related to logistic regression. During training, labeled examples $\{(x_i, y_i)\}_i$ become TensorLog examples of the form `predict(`$x_i, y_i$`)`.

## 2.2 Declaratively Describing Entropic Regularization

A commonly used approach to SSL is entropy regularization, where one introduces a regularizer that encourages the classifier to predict some class confidently on unlabeled examples. Entropic regularizers encode a common bias of SSL systems: the decision boundaries should be drawn in low-probability regions of the space. For instance, transductive SVMs maximize the "unlabeled data margin" based on the low-density separation assumption that a good decision hyperplane lies on a sparse area of the feature space [13].

To implement entropic regularization, we extend TensorLog's interpreter to support a new predicate, `entropy(Y,H)`, which is defined algorithmically, instead of by a set of database facts. It takes the distribution of variable $Y$ as input, and outputs a weighting over the two entities `high` and `low` such that `high` has weight $S_Y$ and `low` has weight $1 - S_Y$, where $S_Y$ is the entropy of the distribution of values $Y$. (In the experiments we actually use Tsallis entropy with $q = 2$ [18], a variant of the usual Boltzmann–Gibbs entropy). Using this extension, one can implement entropic regularization SSL by adding a single rule to the theory of Figure 1. These *entropic regularization (ER) rules* are shown in Figure 2.

In learning, each unlabeled $x_i$ is converted to a training example of the form `predictionHas-Entropy(`$x_i$`,low)`, which thus encourages the classifier to have low entropy over the distribution of predicted labels for $x_i$. During gradient descent, parameters are optimized to maximize the probability of `low`, and thus minimize the entropy of $Y$.

## 2.3 Declaratively Describing a Co-training Heuristic

Another popular SSL method is co-training [6], which is useful when there are two independent feature sets for representing examples (e.g., words on a web page and hyperlinks into a web page). In co-training, two classifiers are learned, one using each feature set, and an iterative training process is used to ensure that the two classifiers agree on unlabeled examples.

**Supervised Classification**
predict(X,Y) ← hasFeature(X,F),indicates(F,Y).

**Entropic Regularization (ER)**
predictionHasEntropy(X,H) ← predict(X,Y), entropy(Y,H).

**Co-training (CT)**
predictionHasEntropy(X,H) ← predict(X,Y), entropy(Y,H).
predict(X,Y) ← predict1(X,Y).
predict(X,Y) ← predict2(X,Y).
predict1(X,Y) ← hasFeature1(X,F), indicates1(F,Y).
predict2(X,Y) ← hasFeature2(X,F), indicates2(F,Y).

**Neighbor Entropy Regularization (NBER)**
neighborPredictionsHaveEntropy(X1,H) ← near(X1,X2), predict(X2,Y2), entropy(Y2,H).

**Label-Propagation Entropy Regularization (LPER)**
nearbyPredictionsHaveEntropy(X1,H) ← sim(X1,X3), predict(X3,Y3), entropy(Y3,H).
sim(X1,X3) ← near(X1,X3).
sim(X1,X3) ← near(X1,X2), sim(X2,X3).

**Co-linked Label Propagation Entropy Regularization (COLPER)**
nearbyPredictionsHaveEntropy(X1,H) ← sim(X1,X3), predict(X3,Y3), entropy(Y3,H).
sim(X1,X3) ← near(X1,X3).
sim(X1,X3) ← near(X1,Z), near(Z,X2), sim(X2, X3).

Figure 2: Declaratively specified SSL rules

Here we instead use an entropic constraint to encourage agreement. Specifically, we construct a theory that says that if the predictions of the two classifiers are disjunctively combined (on an unlabeled example), then the resulting combination of predictions should have low entropy. These are shown as the *co-training (CT) rules* in Figure 2.

The same types of examples would be provided as above: the predict($x_i,y_i$) examples for labeled ($x_i, y_i$) pairs would encourage both classifiers to classify the labeled data correctly, and the examples predictionHasEntropy($x_j$,low) for unlabeled $x_j$ would encourage agreement.

## 2.4 Declaratively Describing Network Classifiers

Another common setting in which SSL is useful is when there is some sort of neighborhood structure that indicates pairs of examples that are likely to have the same class. An example of this setting is hypertext categorization, where two documents are considered to be a pair if one cites another. If we assume that a hyperlink from document $x_1$ to $x_2$ is indicated by the fact near(x1,x2) then we can encourage a classifier to make similar decisions for the neighbors of a document with the *neighbor entropy regularization (NBER) rules* in Figure 2. Here the unlabeled examples would be converted to TensorLog examples of the form neighborPredictionsHaveEntropy($x_j$,low).

Variants of this SSL algorithm can be encoded by replacing the near conditions with alternative notions of similarity. For example, another way in which links between unlabeled examples are often used in *label propagation* methods, such as harmonic fields [31] or modified absorption [25]. If the weights of near facts are less than one, we can define a variant of random-walk proximity in a graph with recursion, using the rule *label-propagation entropy regularization (LPER)* (plus the usual "predict" rule), as shown in Figure 2. Note that in this SSL model, we are regularizing the feature-based classifier to behave similarly to a label propagation learner, so the learner is still inductive, not transductive.

Another variant of this model is formed by noting that in some situations, direct links may indicate different classes: e.g., label propagation using links for an "X1 / X2 is the advisor of Z" relationship may not be good at predicting the class labels "professor" vs "student". In many of these cases, replacing the condition near(X1,X2) with a two-step similarity near(X1,Z),near(Z,X2), improves label propagation performance: in the case above, labels would be propagated through a "X1 co-advises a student with X2" relationship, and in the case of hyperlinks, the relationship would be a co-citation relationship. Below we will call this the *co-linked label propagation entropy regularization (COLPER) rules*, as shown in Figure 2.

## 3 Experimental Results – Text Categorization

Following [3], we consider SSL performance on three widely-used benchmark datasets for classification of hyperlinked text: Citeseer, Cora, and PubMed [21]. We apply one rule for entropic regularization (ER) and three rules for network classifiers (NBER, LPER, COLPER). For this task, the co-training heuristic is not applicable.

## 3.1 The Task and Model Configuration

Many datasets contain data that are interlinked. For example, in Citeseer, papers that cite each other are likely to have similar topics. Many algorithms are proposed to exploit such link structure to

improve the prediction accuracy. In this experiment, the objective is to classify documents, given a bag of words as features, and citation relation as links between them.

An described above, we use `hasFeature(`$d_i$`, `$w_j$`)` as a fact if word $w_j$ exists in document $d_i$ and use this to declaratively specify a classifier. For unlabeled data, we add entropic regularization (ER) by adding a `predictionHasEntropy(`$d_i$`, low)` training example for all unlabeled documents $d_i$. To apply the rules for network classifiers in Section 2.4, we consider `near(X1, X2)` as a citation relation, i.e. if document $d_i$ cites document $d_j$, then `near(`$d_i$`, `$d_j$`)` and `near(`$d_j$`, `$d_i$`)`. We also define every document to be "near" itself, so `near(`$d_i$`, `$d_i$`)` is always true. Then, we can simply apply the NBER, LPER, and COLPER rules to every unlabeled document.

During training, we have five losses: supervised classification, ER, NBER, LPER, and COLPER. These are combined with different weights of importance:

$$l_{total} = l_{predict} + w_1 \cdot l_{ER} + w_2 \cdot l_{NBER} + w_3 \cdot l_{LPER} + w_4 \cdot l_{COLPER}$$

where $w_i$'s are hyper-parameters that will be tuned with Bayesian Optimization [22] [2].

In this experiment, our learning algorithm is closely related to logistic regression. However, since the SSL strategies are based the predicate `predict(X, Y)`, *one could replace this learner with any other learning algorithm*–in particular one could replace it with a non-declarative specified prediction rule as well, using the same programming mechanism we used to define entropy. Exactly the same rules could be used to define the SSL strategies.

## 3.2 Results

We take 20 labeled examples for each class as training data, and reserve 1,000 examples as test data. Other examples are treated as unlabeled. We compare our results with baseline models from D-Learner [3]: supervised SVM with linear kernel (SL-SVM), supervised version of ProPPR (SL-ProPPR) [27], and the D-Learner. Our model variants show consistent improvement over baseline models, as shown in table 1. More importantly, our full model, i.e. "All", performs the best, which shows combining constraints can further improve performance.

Table 1: Text categorization results in percentage (Note that table 1a and 1b use different data splits.)

(a) Results of using different rules

|  | CiteSeer | Cora | PubMed |
|---|---|---|---|
| SL-SVM | 55.8 | 52.0 | 66.5 |
| SL-ProPPR | 52.8 | 55.1 | 68.8 |
| D-Learner | 55.1 | 58.1 | 69.9 |
| DCE (Ours): | | | |
| Supervised | 59.8 | 59.3 | 71.8 |
| + ER | 60.3 | 59.9 | 72.7 |
| + NBER | 61.4 | 60.3 | 72.5 |
| + LPER | 61.3 | **60.5** | 73.1 |
| + COLPER | 60.9 | 60.2 | 73.3 |
| + All | **61.7** | **60.5** | **73.8** |

(b) Results compared with other models
(Our model is inductive, not transductive)

|  | CiteSeer | Cora | PubMed |
|---|---|---|---|
| SL-logit | 57.2 | 57.4 | 69.8 |
| SemiEmb | 59.6 | 59.0 | 71.1 |
| ManiReg | 60.1 | 59.5 | 70.7 |
| GraphEmb | 43.2 | **67.2** | 65.3 |
| Planetoid-I | 64.7 | 61.2 | **77.2** |
| DCE (Ours): | | | |
| Supervised | 63.6 | 60.7 | 72.7 |
| + All | **65.7** | 61.5 | 74.4 |
| Transductive: | | | |
| TSVM* | 64.0 | 57.5 | 62.2 |
| Planetoid-T* | 62.9 | 75.7 | 75.7 |
| GAT* | 72.5 | 83.0 | 79.0 |

We also compare our model with several other models[3]: supervised logistic regression (SL-Logit), semi-supervised embedding (SemiEmb) [28], manifold regularization (ManiReg) [1], TSVM [13], graph embeddings (GraphEmb) [17], and the inductive version of Planetoid (Planetoid-I) [29]. The DCE-Learner performs better on the smaller datasets, Citeseer and Cora, and Planetoid-I (with a more complex multi-layer classifier) performs better on PubMed.

The models compared here, like the DCE-Learner, are inductive: they do not use any graph information at classification time, nor to they access the test data at training time. Inductive learners are

[2] Open source Bayesian optimization tool available at `https://github.com/HIPS/Spearmint`
[3] Data splits available at `https://github.com/kimiyoung/planetoid`

much more efficient at test time; however, it is worth noting that better accuracy can often be obtained by transductive methods. For reference we also give results for TSVM, the transductive version of Planetoid (Planetoid-T), and a recent variant of graph convolutional networks, Graph Attention Networks (GAT) [26], which to our knowledge is the current state-of-the-art on these techniques.

## 4 Experimental Results – Relation Extraction

Another common task in NLP is relation extraction. In this experiment, we start with two distantly supervised information extraction pipelines, DIEL [2] and DIEJOB [4], which extract relation and type examples from entity-centric documents. Then, we train our classifier with several declaratively defined constraints, including one rule for co-training heuristic (CT) and a few variants of constraints for network classifiers (NBER and COLPER). Experimental results show our model consistently improves the performance on two datasets in drug and disease domains.

### 4.1 The Task and Data Preparation

In an entity-centric corpus, each document describes some aspects of a particular entity, called the subject entity. For example, the Wikipedia page about "Aspirin" is an entity-centric document, which introduces its medical use, side effects, and other information. The goal of this task is to predict the relation of a noun phrase to its subject. For example, we would like to determine if "heartburn" is a "side effect" of "Aspirin". Since the subjects of documents are simply their titles in an entity-centric corpus, the task is reduced to classifying a noun phrase X into one of several pre-defined relations R, i.e. predictR(X,R).

We ran experiments on two datasets in the drug and disease domains, respectively: DailyMed with 28,590 articles and WikiDisease with 8,596 articles. These datasets are described in [5]. We directly employ the preprocessed corpora from [5] [4], which contains shallow features such as tokens from the sentence containing the noun phrase and unigram/bigrams from a window around it, and also features derived from dependency parses. In the drug domain, we predict if a noun phrase describes one of the three relations: "side effect", "used to treat", and "conditions this may prevent". If none of the above relation is true, a noun phrase should be classified as "other". In the disease domain, we predict five relations: "has treatment", "has symptom", "has risk factor", "has cause", and "has prevention factor".

Following prior work, labels were produced using DIEJOB [4] to extract noun phrases that have non-trivial relations to their subjects. Since a noun phrase could have different meanings under different context, each mention is treated independently. DIEJOB employs a small but well-structured corpus to collect some confident examples with distant supervision as seeds, then propagates labels in a bipartite graph (links are between noun phrases and their features) to collect a larger set of training examples. Then, we use DIEL [2] to extract predicted types of noun phrases, where types indicate the ontological category of the referent, rather than its relationship to the subject entity. There are six pre-defined types: "risk", "symptom", "prevention", "cause", "treatment", and "disease". It's obvious that types and relations are related, so we'll use it to build our co-training heuristics later. DIEL uses coordinate lists to build a bipartite graph, and starts to propagate with type seeds to collect a larger set of type examples as training data.

### 4.2 Model Configuration

We then introduce a domain-specific set of SSL constraints for this problem. The most confident outputs from DIEL and DIEJOB are selected as distantly labeled examples. However, some examples could be misclassified, since both models are based on label propagation with a limited number of seeds. To mitigate this issue, and to exploit unlabeled data, we design a variant of the co-training constraint (CT) to merge the information from relation and type. Relations and types are connected with a predicate hasType(R, T), which contains four facts in the drug domain, as shown in Figure 3. For example, the third one fact says: the tail entity of "side effect" relation should be of "symptom" type. For disease domain, we define such facts similarly.

**Co-training(CT)**
hasType(conditions_this_may_prevent, disease)
hasType(used_to_treat, disease)
hasType(side_effects, symptom)
hasType(other, other)

predictionHasEntropy(X,H) ← predict(X,T), entropy(T,H)
predict(X,T) ← predictT(X,T)
predict(X,T) ← predictR(X,R), hasType(R,T)

**Neighbor Entropy Regularization (NBER)**
pairPredictionsHaveEntropy(P,H) ← hasExample(P,X1), predict(X1,Y), entropy(Y,H).

**Co-linked Label Propagation**
**Entropy Regularization (COLPER)**
setPredictionsHaveEntropy(P,H) ← hasExampleSet(P,X2), predict(X2,Y), entropy(Y,H).
hasExampleSet(P,X2) ← hasExample(P,X2)
hasExampleSet(P,X2) ← hasExample(P,X1), inPair(X1,P2), hasExampleSet(P2, X2)

Figure 3: SSL rules for relation extraction

We could easily get two classifiers to predict the type and relation of a noun phrase: `predictR(X,R)` and `predictT(X,T)`, which are simple classifiers as described in Section 2.1 using their own features. Then, `hasType(R, T)` converts its relation R to type T. Following the co-training rule (CT) in Section 2.3, `predictionHasEntropy(`$x_i$`, low)` (softly) forces two predictions to match for each unlabeled noun phrase $x_i$, as shown in Figure 3.

In addition to the co-training constraint (CT), we construct another constraint with the assumption that a noun phrase mentioned multiple times in the same document should have the same relationship to the subject entity. For example, if "heartburn" appears twice in the "Aspirin" document, they should both be labeled as "side effect". As a common fact, "heartburn" can't be a "symptom" and a "side effect" of a specific drug at the same time. This constraint could be implemented as a variant of neighbour entropy regularization (NBER). Let $x_i$ and $x_j$ be two mentions of a noun phrase, and $p_{x_i x_j}$ is a virtual entity that represents the pair $(x_i, x_j)$. Conceptually, $p_{x_i x_j}$ is a parent node, and $x_i$ and $x_j$ are its children. In analogy to `near(X1,X2)`, we create a new predicate `hasExample(P,X)` and consider `hasExample(`$p_{x_i x_j}$`, `$x_i$`)` and `hasExample(`$p_{x_i x_j}$`, `$x_j$`)` as facts. Now, we are ready to create a training example `pairPredictionsHaveEntropy(`$p_{x_i x_j}$`, low)`, which encourages $x_i$ and $x_j$ to be classified into the same category. This NBER constraint is shown in Figure 3.

It is obvious that $x_i$ and $x_j$ in the previous example are of distance two, i.e. $x_i$ and $x_j$ "co-advise" $p_{x_i x_j}$. Inspired by co-linked label propagation entropy regularization (COLPER), we could recursively force a set of examples to have the same prediction. `hasExampleSet(P,X)` is recursively defined as shown in Figure 3, where `inPair(X,P)` is symmetric to `hasExample(P,X)`, i.e. `inPair(`$x_i$`, `$p_{x_i x_j}$`)` iff `hasExample(`$p_{x_i x_j}$`, `$x_i$`)`. Using `hasExampleSet(P,X)`, we expand a pair of noun phrases $(x_i, x_j)$ to a set $\{x_1 \cdots x_n\}$ in which any pair of noun phrases are similar.

Another observation is that a noun phrase in the same section always has a similar relationship to its subject, even across different documents. For example, if "heartburn" appears in the "Adverse reactions" section of "Aspirin" and "Singulair", both mentions should both be "side effect", or else neither shuld be. Given a pair of mentions from the sections with the same name, we also construct training examples for the NBER rule `pairPredictionsHaveEntropy(E,H)`. Note that we use the same rules, but examples are prepared under different assumptions.

Similar to text categorization, for each unlabeled example $x_i$, we also add an ER example `predictionHasEntropy(`$x_i$`, low)`. Rules are combined with a weighted sum of loss, and weights are tuned with Bayesian Optimization.

## 4.3   Results

In the drug domain, we take 500 relation and type examples for each class as labeled data and randomly select 2,000 unlabeled examples for each constraint. In the disease domain, we take 2,000 labeled relation and type examples for each class and 4,000 unlabeled examples for each constraint.

The evaluation dataset was originally prepared in DIEJOB, which contains 436 and 320 examples for the disease and drug domains. The model is evaluated from an information retrieval perspective. We predict the relation of all noun phrases in test documents and drop those that are predicted as "other". Then, we compare the retrieved examples with the ground truth to calculate the precision, recall and F1 score. There is also a tuning dataset, please refer to [4] for more details of evaluation, tuning data, and the evaluation protocols.

Besides D-Learner, we compare our experiment results with the following baseline models[5]. The first four are supervised learning baselines via distant supervision: DS-SVM and DS-ProPPR directly

Table 2: Relation extraction results in F1

(a) Results compared to baseline models

| | Disease | Drug |
|---|---|---|
| DS-SVM | 27.1 | 17.8 |
| DS-ProPPR | 21.9 | 17.2 |
| DS-Dist-SVM | 27.5 | 30.0 |
| DS-Dist-ProPPR | 24.3 | 25.3 |
| MultiR | 24.9 | 14.6 |
| Mintz++ | 24.9 | 17.8 |
| MIML-RE | 26.6 | 16.3 |
| D-Learner | 31.6 | 37.8 |
| DCE (Ours): | **33.3** | **50.1** |

(b) Results of using different rules

| | Disease | Drug |
|---|---|---|
| DCE (Supervised) | 30.4 | 46.6 |
| + ER-Relation | 31.4 | 48.5 |
| + ER-Type | 31.2 | 48.3 |
| + NBER-Doc | 32.5 | 48.2 |
| + NBER-Sec | 32.4 | 47.9 |
| + COLPER-Doc | 32.4 | 48.4 |
| + COLPER-Sec | 31.9 | 48.7 |
| + CT | 31.2 | 49.8 |
| + All | **33.3** | **50.1** |

employ the distantly labeled examples as training data, and use SVM or ProPPR as the learner, while DS-Dist-SVM and DS-Dist-ProPPR first employ DIEJOB to distill the distant examples, as done for D-Learner and our model. The second group of comparisons include three existing methods: *MultiR*, [12] which models each mention separately and aggregates their labels using a deterministic OR; *Mintz++* [23], which improves the original model [16] by training multiple classifiers, and allowing multiple labels per entity pair; and *MIML-RE* [23] which has a similar structure to *MultiR*, but uses a classifier to aggregate mention level predictions into an entity pair prediction. We use the public code from the authors for the experiments[6].

Results in Table 2 show that entropic regularizations (ER, NBER, and COLPER) and co-training heuristics (CT) do improve the performance of the model. Our full model achieves a new state-of-the-art result. We observe that different rules yield various improvement on performance. Furthermore, a weighted combination of rules can improve the overall performance, which again justifies that it is an appropriate approach to combine constraints.

# 5  Related Work

This work builds on Bing et al. [3], who proposed a declarative language for specifying semi-supervised learners, the *D-Learner*. The DCE-Learner explored here has similar goals, but is paired with a more effective and more flexible underlying learning system, and experimentally improves over the original D-Learner on all our benchmark tasks, even though its constraint language is somewhat less expressive.

Many distinct heuristics for SSL have been proposed in the past (some of which we discussed above, e.g., making confident predictions at unlabeled points [13]; imposing consistency constraints based on an ontology of types and relations [7]; instances associated with the endpoints of an edge having similar labels [31, 1, 25] or embedded representations [28, 30, 14]). Some heuristics have been formulated in a fairly general way, notably the information regularization of [24], where arbitrary coupling constraints can be used. However, information regularization does not allow multiple coupling types to be combined, nor does have an obvious extension to support label-propagation type regularizers, as the DCE-Learner does.

The regularizers introduced for relation extraction are also related to well-studied SSL methods; for instance, the rules encouraging agreement between predictions based on the type and relationship are inspired by constraints used in NELL [7, 19], and the remaining relation-extraction constraints are specific variants of the "one sense per discourse" constraint of [10]. There is also a long tradition of incorporating constraints or heuristic biases in distant-supervision systems for relation extraction, such as DIEBOLD [2], DIEJOB [5], MultiR [12] and MIML [23]. The main contribution of the DCE-Learner over this prior work is to provide a convenient framework for combining such heuristics, and exploring new variants of existing heuristics.

The Snorkel system [20] focuses on declarative specification of sources of weakly labeled data; in contrast we specify entropic constraints over declarative-specified groups of examples. Previous work has also considered declarative specifications of agreement constraints among classifiers (e.g., [19]) or strategies for a classification learning [9] and recommendation (e.g., [15]. However, these prior systems have not considered declarative specification of SSL constraints. We note that constraints are most useful when labeled data is limited, and SSL is often applicable in such situations.

## 6 Conclusions

SSL is a powerful method for language tasks, for which unlabeled data is usually plentiful. However, it is usually difficult to predict which SSL methods will work well on a new task, and it is also inconvenient to test alternative SSL methods, combine methods, or develop novel ones. To address this problem we propose the DCE-Learner, which allows one to declaratively state many SSL heuristics, and also allows one to combine and ensemble together SSL heuristics using Bayesian optimization. We show consistent improvements over an earlier system with similar technical goals, and a new state-of-the-art result on a difficult relation extraction task.

## Footnotes

[4]Data available at `http://www.cs.cmu.edu/~lbing/#data_aaai-2016_diebolds`

[5]Refer to [3] for the full details.

[6]Code available at `http://aiweb.cs.washington.edu/ai/raphaelh/mr/` and `http://nlp.stanford.edu/software/mimlre.shtml`

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
