[Reviews · NeurIPS 2018]

Reviewer 1



This paper proposes a declarative specifications for semi-supervised learning (SSL) and specifies various different SSL approaches in this framework. The experiments on document classification and relationship extraction tasks show improvements compared to other methods. I like the approach as the paper improves on a previous method D-Learner and is able to combine different SSL strategies, e.g. co-training, neighbourhood based label propagation, ... using adaptations to entropy regularisation rule. It also shows good experimental performance on smaller datasets. Where I think the paper can improve is to experiment on text classification or RE datasets that are considerably larger than the ones studied. These datasets provide possibilities of comparison to simpler methods that rely on large unlabelled datasets. For example for RE there're models such as "Neural Relation Extraction with Selective Attention over Instances", Lin et al.,2016 or similar techniques rely on full weak supervision rather than SSL. For text classification, large scale unsupervised methods with pre-trained language models are often seen to perform really well with small number of labeled examples ("Learning to Generate Reviews and Discovering Sentiment", Radford et al 2017, "Improving Language Understanding by Generative Pre-Training", Radford et al 2018).

Reviewer 2



The submission introduces a variant of the D-Learner declarative language, which is limited to learners based on ProPPR probabilistic logic. The proposed variant, DCE-Learner is based in TensorLog which, while a more constrained language, has the benefit of a more extensive library of learners, among other things. A set of declarative rules are specified to encode different SSL settings. Experimental results are given showing improved performance on text classification and entity extraction. Unfortunately, for someone unfamiliar with declarative languages, it's difficult to distinguish how novel these rules are, given the TensorLog logic. The second experiment appears to be where the methods are most effective, but the limited space allotted to describing the problem setting and methodology makes the logic difficult to follow. The paper might be stronger if more room was given either to how TensorLog implements these rules or to the relation extraction task. - in Figure 2, what is the difference between near(.,.) and sim(.,.)? - in Table 1, doesn't GraphEmb perform better than DCE on the Cora dataset? Is that an error in the table, or in line 178? - in Table 1(a), is there an intuition as to why the base supervised DCE outperforms the othe other learners? Is this simply an effect of the TensorLog implementation or is there something about the design of the DCE that contributes to performance? - In Table 1, why do the DSE Supervised and All results differ between (a) and (b)? - in section 4.1, is it possible to more explicitly state the difference between a 'relation' and a 'type'? - how important is the Bayesian optimization used to combine the SSL ensemble? Are there any hyperparameters to tune here? ---------------------- Thank you to the author's for their clear response. I feel they have done a good job of addressing any issues raised and propose changes which will make the paper clearer.

Reviewer 3



This paper proposes a method to combine (or ensemble) several SSL heuristics (regularizers) by using a Bayesian optimization approach. The basic idea of the proposed method borrowed from the previous method called D-Learner, which is declared in this paper. Therefore, the proposed method is basically a modification or extension of D-Learner, which seems not to be totally novel. In this perspective, this paper is rather incremental than innovative. The experimental results look fairly well comparing with the methods in previous studies including the baseline D-Learner on the tasks of text classification and relation extraction examined in this paper. The followings are my main concerns of this paper. If I misunderstand something, please provide rebuttals to my concerns. I am willing to change my score if I think the authors’ rebuttal is reasonable. 1, The proposed method seems to deeply depend on the TensorLog framework. From this perspective, I somehow feel that this paper is rather a technical report than a scientific paper. I would like authors to clearly explain that what the central theory of the proposed method is, or what the unique idea of the proposed method is. For example, the idea of incorporating many SSL regularizers together in a single leaner is an idea of the previous study. 2, For example, the DCE(Supervised) in Table 2(b), which is the baseline of the proposed method, already outperformed the performance for all the methods in Table 2(a). I wonder why the baseline of the proposed method got such higher performance. From this point, I concern the experimental settings used in this paper whether the comparison was performed in a fair condition. 3, The results for DCE (Ours) shown in Tables 1 (a) and (b) differs. I cannot figure out why this difference was induced from. Please clearly explain this. After reading author's feedback: Thank you for answering my concerns/questions. Unfortunately, I am still not totally convinced the novelty of the proposed method. Moreover, the reason of the supervised DCE-Learner has already outperformed the most of previous methods is still not super clear for me. Is there any particular reason why any of previous methods did not use SL DCE-Learner (or similar strong baseline SL methods)? It is a bit hard for me to "strongly" recommend this paper for the acceptance. In addition, I would recommend authors to report the results of both splits to prevent the misunderstanding in the next version.